

# Soil microbial community structure and diversity are largely influenced by soil pH and nutrient quality in 78-year-old tree plantations

Xiaoqi Zhou[1,2,4], Zhiying Guo[3], Chengrong Chen[2], Zhongjun Jia[3]

[1]Center for Global Change and Ecological Forecasting, Tiantong National Forest Ecosystem Observation and Research Station, East China Normal University, Shanghai 200241, China

[2]Australian Rivers Institute and Griffith School of Environment, Griffith University, Nathan, Brisbane 4111, Queensland, Australia

[3]State Key Laboratory of Soil and Sustainable Agriculture, Institute of Soil Science, Chinese Academy of Sciences, Nanjing, Jiangsu Province 210008, China

[4]Shanghai Key Lab for Urban Ecological Processes and Eco-restoration, School of Ecological and Environmental Sciences, East China Normal University, Shanghai 200241, China

*Correspondence to*: Zhongjun Jia (jia@issas.ac.cn)

**Abstract.** Forest plantations have been recognized as a key strategy management tool for stocking carbon (C) in soils, thereby contributing to climate warming mitigation. However, long-term ecological consequences of anthropogenic forest plantations on the community structure and diversity of soil microorganisms and the underlying mechanisms determining these patterns are poorly understood. In this study, we selected 78-year-old tree plantations that included three coniferous tree species (i.e., slash pine, hoop pine and kauri pine) and an eucalypt species in subtropical Australia. We investigated the patterns of community structure, and the diversity of soil bacteria and eukaryotes by using high-throughput sequencing of 16S rRNA and 18S rRNA genes. We also measured the potential methane oxidation capacity under different tree species. The results showed that slash pine and *Eucalyptus* significantly increased the dominant taxa of bacterial Acidobacteria and the dominant taxa of eukaryotic Ascomycota, and formed clusters of soil bacterial and eukaryotic communities, which were clearly different from the clusters under hoop pine and kauri pine. Soil pH and nutrient quality indicators such as C:nitrogen(N) and extractable organic C:extractable organic N were key factors determining the patterns of soil bacterial and eukaryotic communities among the different tree species treatments. Slash pine and *Eucalyptus* had significantly lower soil bacterial and eukaryotic operational taxonomical unit numbers and lower diversity indices than kauri pine and hoop pine. A key factor limitation hypothesis was introduced, which gives a reasonable explanation for lower diversity indices under slash pine and *Eucalyptus*. In addition, slash pine and *Eucalyptus* had a higher soil methane oxidation capacity than the other tree species. These results suggest that significant changes in soil microbial communities may occur in response to chronic





disturbance by tree plantations, and highlight the importance of soil pH and physiochemical characteristics in microbially-mediated ecological processes in forested soils.

## 1 Introduction

Forest plantations are widely used to mitigate carbon dioxide ($CO_2$) concentrations in the atmosphere (Stocker et al., 2013). Worldwide, about 68,000 ha of non-forest land have been planted to forest during the past 15 years, which has been recognized as a management tool for stocking carbon (C) in soils, thereby contributing toward climate change mitigation (Keenan et al., 2015). Afforestation with different tree species has been accepted as an effective measure for increasing soil

C stocks. However, considering the demand for timber of certain tree species and to optimize the productivity of stemwood, forest farmers usually select certain tree species such as coniferous tree species (Lu et al., 2012). The effects of afforestation with different tree species on soil C stocks have been widely studied (Nave et al., 2013; Vesterdal et al., 2013), showing that different tree species can greatly affect soil C sequestration (Vesterdal et al., 2013). However, the community assembly of complex soil microorganisms remains largely unexplored at a fine taxonomic resolution, although these microorganisms

catalyze soil functional processes in close association with soil C transformation.

  A large fraction of soil C and nitrogen (N) contents are arguably derived from tree litter, and thus differences in the chemistry of plant traits may be a key driving force in shaping the abundance and composition of soil bacterial and fungal communities to various extents (Prescott and Grayston, 2013). In fact, strong correlations between microbial communities and dominant trees has been observed, implying that the composition of fungal and bacterial communities in both soil and

forest litter is largely determined by the tree plantation (Urbanová et al., 2015). In turn, microbial communities are responsible for organic matter mineralization and providing nutrients for tree growth, and are thus an integral component of global C and N cycles. Generally, soil with higher C:N ratios under different tree species has more recalcitrant biopolymers, which are represented by the polysaccharides and cellulose that are used by decomposer microorganisms. Because of their filamentous form, fungi tend to be more involved in the decomposition of polymeric compounds (de Boer et al., 2005).

Bacteria that preferentially use organic compounds with a low molecular mass may rely on the products of fungal–biopolymer decomposition for nutrition (Tedersoo et al., 2008). Soil with lower C:N ratios, which tend to have more organic compounds with a low molecular mass has more bacteria than fungi. Although much research has focused on the effects of tree plantations on soil C storage, as an overall measure for soil ecosystem services, the ecologically important processes of microbial functional guilds are rarely investigated because of the low resolution of conventional culture-dependent

techniques.

  Upland forest soils mostly function as significant methane ($CH_4$) sinks (Kolb, 2009) and $CH_4$ oxidation capacity in these soils is sensitive to land uses such as afforestation with different tree species (Tate, 2015). Soil $CH_4$-oxidizing bacteria are considered to be the sole catalyst for consumption of $CH_4$ in the atmosphere by forest soils (Knief et al., 2005). Intriguingly, the diversity of $CH_4$-oxidizing bacteria appears to explain the $CH_4$ oxidation capacity in grassland ecosystems (Zhou et al.,





2008) or in forest ecosystems (Levine et al., 2011). However, these studies are largely based on the correlation of flux measurements to functional biomarker *pmoA* genes encoding particulate $CH_4$ monooxygenase in various soils, and these generalized conclusions warrant further investigation. Furthermore, the underlying mechanisms of how abiotic factors select for specific microbial taxa at different taxonomic ranks remains largely uncertain.

Previous studies have shown that the community structure and diversity of soil microorganisms are mainly influenced by soil pH (Rousk et al., 2010), soil organic C (Sul et al., 2013) and/or soil nutrient concentrations (Lauber et al., 2008). Previously we introduced the key factor limitation hypothesis to explain the differences in soil microbial diversity and found that in a given habitat, when a certain key influencing factor becomes more and more constrained, microbial diversity will increase (Zhou et al., 2011). Here, we selected a 78-year-old tree plantations that was developed on the same soil parent

material in order to test whether soil microbial diversity increased or not under progressively limited conditions of key physiochemical factors resulting from different tree species regimes.

Based on these hypothesis, the objectives of this study were to (1) investigate the selective effects of long-term tree plantations on the community assembly of soil bacteria and eukaryotes; (2) elucidate the functional efficiency of soil $CH_4$ oxidation processes and $CH_4$-oxidizing bacterial communities under different tree species; and (3) determine the key factors

influencing the community structure and diversity of soil bacteria and eukaryotes under different tree species.

## 2 Materials and methods

### 2.1 Experimental site

We selected a 78-year-old forest plantation with different tree species which was established in 1935 on a site that was originally a banana farm. The forest plantation site is located at Cooloola, Tin Can Bay, Southeast Queensland, Australia ($25^o56'49''$S, $153^o5'27''$E). The altitude is 43 m above sea level with a mean annual rainfall of 1287 mm. Winter temperatures range from $7^o$C to $23^o$C over June to August and summer temperatures range from $18^o$C to $30^o$C over December to February

(Lu et al., 2012). Four tree species were selected, including an exotic coniferous species (slash pine (*Pinus elliottii* Engelm. Var. elliottii)) and two native conifer species (hoop pine (*Araucaria cunninghamii* Ait) and kauri pine (*Agathis robusta* C. Moore)), as well as a *Eucalyptus* species (*Eucalyptus grandis* W Hill ex Maiden). All of them were planted adjacently on a broad, gently undulating plain with a gentle slope of less than $5^o$. The plot size of each tree species was 1.087, 0.308, 0.428 and 0.60 ha, respectively (Lu et al., 2012). Four subplots of 10 m × 20 m in each tree plantation were randomly selected for

soil sampling, resulting in a total of 16 subplots. The thicknesses of the litter and fermentation layers were 5–6 cm and 1–2 cm for slash pine, respectively, whereas the corresponding values were 4–5 cm and 1–2 cm for the hoop pine and kauri pine plots. The *Eucalyptus* plot had a thicker litter layer of 8–10 cm and a similarly thick fermentation layer of 1–2 cm.

### 2.2 Soil sampling and measurement of soil physiochemical properties





Soil samples were collected in August 2013 using a diagonal sampling pattern (i.e., one point at each corner and one in the center of each plot) using a soil auger (8-cm in diameter) at 0–10 cm depth within each plot. The soil cores were immediately mixed thoroughly and kept in a cooler (4 $^{\circ}$C). After passing the samples through a 2-mm sieve to remove roots

and stones, the soil samples were stored at 4 $^{\circ}$C prior to analysis. Part of each fresh sample was stored at 4$^{\circ}$C for analysis of soil moisture and pH. The other parts were air-dried and stored at room temperature for soil extractable organic C (EOC) and N (EON) analysis via hot water extraction (Zhou et al., 2013b). After air-dried soil samples were finely ground, soil total C and N contents and $\delta^{15}$N were determined using an Isoprime isotope ratio mass spectrometer with a Eurovector elemental analyzer (Isoprime-EuroEA 3000) (Zhou et al., 2012). Soil moisture content was determined after samples were oven-dried

at 105$^{\circ}$C overnight. The particle size of these soils was dominated by the sand fraction (~96%). All soil biochemical properties are shown in Table 1.

### 2.3 Soil genomic DNA extraction and amplification

Nucleic acids were extracted from 0.5 g of each sample by using the procedures described previously elsewhere (Zhou et al., 2010). In brief, soil was placed in a 2-mL screwcap tube containing a mixture of ceramic and silica particles (Bio101, Carlsbad, Calif.); the mixture was homogenized for 30 s in a FastPrep bead beater cell disrupter (Bio101). After the nucleic acids had been precipitated and washed twice in 75% (vol/vol) ethanol, the final DNA was re-suspended in 100 μL double-distilled water. Furthermore, the crude extract was purified with the Qiagen Gel Extraction Kit (Qiagen Inc, Shanghai,

China). DNA concentrations were measured with a Biospec-mini spectrophotometer (Shimadzu, Kyoto, Japan).

The bacterial-specific 16S rRNA primers of 530F and 907R (Poulsen et al., 2013) were used to amplify the bacterial 16S rRNA gene and the eukaryotic-specific primers of Euk528F–Euk706R (Urbanová et al., 2015) were used to amplify the eukaryotic 18S rRNA gene. The final concentration of different components in the polymerase chain reaction (PCR) amplification mixture was as follows: 0.4 μM of each primer, 200 μM of each deoxynucleoside triphosphate, 1.5 mM MgCl$_2$,

1 × thermophilic DNA polymerase and 10 × reaction buffer (MgCl$_2$-free), 1.25 U per 50 μL of Taq DNA polymerase (Promega, Madison, WI), and DNAse- and RNAse- free filter sterilized water. All reactions were performed in a PTC-200 thermal cycler (MJ Research Co., New York, USA). The PCR cycle conditions were: an initial denaturation/activation of the hotstart polymerase at 98 $^{\circ}$C for 30s, followed by 30 cycles of denaturation at 98 $^{\circ}$C for 5s, annealing at 53 $^{\circ}$C for 20s and elongation at 72 $^{\circ}$C for 20s, and then a final elongation step at 72 $^{\circ}$C for 5 min. The PCR products were run on a gel and the

appropriate fragments were cut and purified using Qiagen Gel Extraction kit (Qiagen). The amplicons were sequenced on an Illumina Miseq machine (Illumina, Nanjing, China).



### 2.4 High-throughput sequencing data analysis and statistical analysis

The sequence data were processed mainly using QIIME (Caporaso et al., 2010) and UPARSE (Edgar, 2013). The default filter settings were used: no ambiguous bases were allowed (Huse et al., 2007) and a minimum read length of 150 bp. The raw sequences were demultiplexed by using unique barcodes that allowed no mismatch. Pair-end data of each sample were joined with FLASH (Magoč and Salzberg, 2011) using the default parameters. Reads with average quality scores below 20 were discarded. The high-quality reads were clustered at similarity of 97% to generate operational taxonomic units (OTUs) using UPARSE. Simultaneously, chimeras were filtered. For each OTU, one representative sequence was picked. The Shannon–Wiener index, evenness and chaos scores of soil bacteria and eukaryotes of each tree species were calculated based on OTU data. A taxonomic annotation was assigned to each representative sequence via the assign_taxonomy.py tool in QIIME, using SILVA 123 (Quast et al., 2013) as a reference database. To avoid any potential uncertainty caused by sampling error (Zhou et al., 2013a), each sample was rarefied to the same sequence number: 39,480 reads for 16S rRNA and 28,967 reads for 18S rRNA. The Pearson correlation between soil physiochemical properties and microbial diversity were calculated. All results were tested statistically using ANOVA and Tukey's test, and significance was considered to be at $P<0.05$ among the tree species treatments.

Rarefaction curve were plotted to compare α diversity at different sequencing depths. The Bray–Curtis distance was calculated and principal coordinate analysis (PcoA) was applied to the distance matrix. Canonical correlation analysis (CCA) was used to elucidate the relationship between soil physiochemical properties and the patterns of bacterial and eukaryotic community structure among the tree species treatments. The CCA was performed with the *vegan* package in R software. Structural equation modeling was performed using R software with the *lavaan* package to explore the causal links between soil pH and nutrient quality, as well as soil bacterial and eukaryotic diversities. The model considered soil pH, C:N, EOC:EON and microbial diversity. We considered that path analysis was most appropriate for data sets with large sample sizes but that the samples for the number of variables in every model were too small ($n = 16$). However, small sample sizes generally result in conservative fitting estimates (Shipley, 2000). In structural equation modeling, a $\chi^2$ test is used to determine whether the covariance structures implied by the model adequately fit the actual covariance structures of the data. A non-significant $\chi^2$ test ($P > 0.05$) indicates an adequate model fit. The coefficients of each path, taken as the calculated standardized coefficients, were determined by analyzing the correlation matrices. Paths in this model were considered to be significant at $P<0.05$. These coefficients indicate by how many standard deviations the effect variable would change if the causal variable was changed by one standard deviation (Shipley, 2000).

## 3 Results

### 3.1 Soil bacterial and eukaryotic community structure and diversity



The total number of bacterial sequences and their relative abundance in soils under different tree species are shown in Table 2. Of the total number of sequences from each treatment, only a small percentage (<1%) was unclassified or from Archaeal taxa, whereas the majority (>99%) belonged to bacterial taxa. The bacterial sequences at different phylogenetic levels revealed that the number of orders and families represented an unimodal mode, first increasing and then decreasing with an increasing numbers of sequences, whereas the number of genera shows a relatively flat curve (Fig. S1).

Kauri pine and hoop pine had a significantly higher bacterial Shannon diversity index than slash pine and *Eucalyptus*, whereas kauri pine and hoop pine had significantly higher eukaryotic diversity indices than slash pine, and there were no significant differences in soil eukaryotic diversity index between slash pine and *Eucalyptus* (Table 3). We calculated the mean OTUs for each tree species and the results are shown in Fig. S2. For soil bacterial taxa, kauri pine had the highest bacterial OTUs, followed by hoop pine, while *Eucalyptus* and slash pine had the lowest bacterial OTUs. Kauri pine also had high soil eukaryotic OTUs, followed by hoop pine and *Eucalyptus*, whereas slash pine had the lowest soil eukaryotic OTUs.

Through PcoA analysis of soil bacterial taxa, we found that PcoA1 and PcoA2 explained 61.4% and 9.8% of the variations in soil bacterial communities, respectively, among the tree species treatments (Fig. S3a). Slash pine and *Eucalyptus* formed a cluster, which was clearly separated from the cluster of hoop pine and kauri pine. In addition, PcoA1 and PcoA2 explained 20.5% and 19.1% of the soil eukaryotic community, respectively, among the tree species treatments (Fig. S3b). There were a clear separation among slash pine, kauri pine and *Eucalyptus*, whereas hoop pine did not have a clear separation from the other coniferous tree species.

## 3.2 Relative abundance of soil bacteria and eukaryotes

The relative abundance of specific bacterial taxa in the samples was investigated at different taxonomical levels (phyla and genera). The most frequent phyla for all treatments were Acidobacteria (32–58%), Proteobacteria (19–36%), Actinobacteria (15–19%) and Planctomycetes (5–10%), followed by varying occurrence of *WD272*, Firmicutes, Bacteroidetes, Cyanobacteria, Armatimonadetes and Chloroflexi (Fig. 1). Long-term tree plantations apparently imposed strong selection on the microbial community structure. Slash pine and *Eucalyptus* had significantly higher Acidobacteria, and *WD272*, and markedly higher Actinobacteria proportions than hoop pine and kauri pine, whereas AlphaProteobacteria and Planctomycetes, Bacteroidetes, Armatimonadetes, Chloroflexi, Verrucomicrobia, Gemmatimonadetes and Nitrospirae were apparently stimulated under slash pine and *Eucalyptus* (Fig. 2).

By comparison, eukaryotic taxa in the samples came from five major phyla and eight major genera (Fig. 3). Slash pine and *Eucalyptus* had significantly higher proportions of Ascomycota fungi (59.1–61.6%) than hoop pine (45.0%) and kauri pine (52.1%) (Fig. 3a). *Eucalyptus* had a significantly higher relative abundance of *Leotiomycetes* spp. (22.4%) than the other tree species, whereas slash pine had a significantly higher relative abundance of *Pezizomycotina* spp. (18.9%) than the other tree



species (Fig. 3b). Hoop pine had significantly higher unassigned phyla (23.0%) and unassigned genera (22.9%) than the other three treatments (Fig. 3).

### 3.3 Key factors influencing soil bacterial and eukaryotic communities

CCA eigenvalues indicated that Axes 1 and 2 accounted for 59.2% and 10.3% of the overall variance of soil bacterial communities (Fig. 4a), whereas Axes 1 and 2 explained 20.0% and 15.0% of the overall variance in soil eukaryotic communities (Fig. 4b). The species–environment correlations of soil bacterial and eukaryotic taxa were 0.94 and 0.95, respectively, indicating that soil microbial community patterns were strongly correlated with soil physiochemical properties.

The arrows for soil pH, C:N and EOC:EON were longer than those of the other variables, indicating that these factors accounted for the greatest proportion of the variance in soil bacterial taxa among different tree species treatments. Similarly, soil pH, C:N, EOC:EON and natural $^{15}$N abundance accounted for the greatest proportion of variance in soil eukaryotic community patterns.

The soil bacterial community patterns under slash pine and under *Eucalyptus* were mainly influenced by C:N and
15 EOC:EON, whereas those under hoop pine and under kauri pine were largely influenced by pH and EON. Soil eukaryotic communities under *Eucalyptus* were mainly influenced by TC, TN and EOC, whereas those under hoop pine and under kauri pine were largely influenced by pH and EON (Fig. 4). We calculated the correlation between the key physiochemical properties and the soil bacterial and eukaryotic diversity index, and found that there was a negative relationship between soil pH and the bacterial and eukaryotic diversity indices, whereas soil C:N and EOC:EON had a positive influence on the soil
bacterial and eukaryotic diversity index (Table 4). Through calculating the relative contributions of soil pH and nutrient quality, such as C:N ratios and EOC:EON ratios, to soil bacterial and eukaryotic diversities, we found that soil pH played a predominant role in determining soil microbial diversity (Fig. 5).

### 3.4 Methanotrophic activities and community composition

*Eucalyptus* had the highest potential methanotrophic activity, which was significantly higher than that of hoop pine and kauri pine (Fig. 5). There were no significant differences in potential methanotrophic activity between slash pine and *Eucalyptus*. As revealed by high-throughput sequencing, methanotrophic communities were dominated by members of the *Methylosinus* genus. Slash pine and *Eucalyptus* had significantly higher abundance of *Methylosinus* spp. than hoop pine and kauri pine.
Apart from *Methylosinus*, we also detected other methylotrophic genera: *Methylophilum*, *Microvirga* and *Candidatus methylacidiphilum*. Given that the *Methylosinus* genus is a group of high-affinity methanotrophs that can use $CH_4$ at extremely low levels of appropriately 2.0 parts per million (volume/volume) in the atmosphere (Kolb 2009), they may play an important role as a biological sink of atmospheric $CH_4$ in this region.



## 4 Discussion

Afforestation is a useful management practice for mitigating $CO_2$ emissions derived from anthropogenic activities, and many studies have reported that afforestation can increase soil organic C contents (Nave et al. 2013). In this study, the adjacent

5   plantation forests have relatively flat terrain that was developed from the same parent soil, so the microbial divergence of community assemblies in forest soils was considered to be a net effect resulting from the niche segregation imposed by 78 years of being planted to different tree species. Our results provide compelling evidence for how the microbial community structure depends on soil pH and nutrient quality such as C:N ratios and EOC:EON ratios, which are largely determined by tree plantations.

### 4.1 Community structure and relative abundance of soil bacteria and eukaryotes under tree species

Trees may influence soil microbial communities through direct effects such as root exudation and interactions with root symbiosis and root-associated microorganisms, or through indirect effects such as litter production and alteration of the

microclimate (e.g. temperature and moisture) (Prescott and Grayston, 2013). Regardless of the direct or indirect pathways, the specific tree species plays an important role in determining community structure and the abundance of soil microorganisms (Urbanová et al., 2015). In this study, we found that slash pine and *Eucalyptus* had significantly lower soil pH than hoop pine and kauri pine, which may favor the significantly higher relative abundance of Acidobacteria (Fig. 1). Given that Acidobacteria are considered to be indicators of soil pH status (Lauber et al., 2009), soil pH at the study site

(below 6.5) could be considered acidic (Table 1), which was supported by the dominant Acidobacteria phylum in soil bacterial communities across the tree species treatments (Fig. 1). The relative abundance of Acidobacteria in this study was higher than those in other alkaline soils of terrestrial ecosystems (Rousk et al., 2010; Urbanová et al., 2015).

Similarly, slash pine and *Eucalyptus* seem to favor a higher relative abundance of Ascomycota fungi than hoop pine and kauri pine (Fig. 3), which could be explained by lower soil pH and higher C:N ratios (Fig. 4). Previous studies have stated

that fungal, rather than bacterial, communities are more subject to stronger selection pressure under different tree plantation regimes (de Boer et al., 2005). A reason for this could be that the root-associated filamentous fungi may extend from the tree rhizosphere and carry the legacy of plant traits to the bulk soil. Filamentous fungi are stronger decomposers of litter production with lower nutrient quality and can better cope with environmental stresses such as low pH and a dry environment with limited connectivity (de Boer et al., 2005; Urbanová et al., 2015).

It has been reported that soil microbial communities are strongly influenced by soil pH, soil organic C contents or the C:N ratio in terrestrial ecosystems (Fierer and Jackson, 2006; Fierer et al., 2009; Rousk et al., 2010). However, we found that EOC:EON ratios could also result in the differences in soil bacterial and eukaryotic communities among the tree species treatments (Fig. 4), in addition to soil pH and C:N ratios as previously demonstrated. It thus seems plausible that microorganisms with a specific life strategy for labile C and N use could have been preferentially stimulated. Meanwhile, it




should be emphasized that a large fraction of Acidobacteria remained unculturable, and our results provide strong hints towards substrate optimization to cultivate the as-yet-uncultivated microbes in soils.

### 4.2 Diversity of soil bacteria and eukaryotes under tree species

The negative correlation between the EOC:EON ratio and the microbial diversity index provides further support for our key factor limitation hypothesis, namely that when the key factors influencing soil microbial communities become limiting, microbial diversity will increase (Zhou et al., 2011). A small fraction of microorganisms might have propagated rapidly as a result of the elevation of easily available EOC in slash pine and *Eucalyptus* plantations. This unbalanced growth of microbial

communities thus resulted in significantly lower bacterial OTU numbers, Shannon diversity index and evenness than hoop pine and kauri pine (Table 3). This discrepancy could have resulted from anthropogenic enrichment via these 78-year-old tree plantations.

However, it could not be ruled out that other key soil physiochemical factors may play a substantial role in determining the soil microbial community patterns (Fig. 4). The relationships between soil microbial diversity and these key factors were

15 calculated. We found that nutrient quality indicators such as C:N ratios had significantly negative relationships with the soil bacterial and eukaryotic OTU numbers and Shannon diversity index. EOC:EON ratios had significantly negative relationships with the bacterial OTU numbers and the diversity index, but only had a negative relationship with the eukaryotic OTU numbers and diversity index. This could be attributed to how the dominant eukaryotic taxa were filamentous microorganisms, which can get nutrients from plant roots or other places via hyphae in the heterogeneous

environment. Interestingly, we found that soil pH had significantly positive relationships with the bacterial and eukaryotic OTU numbers and diversity index, which may outweigh the positive role of soil nutrient quality (Fig. 5), resulting in lower soil bacterial and eukaryotic diversity under slash pine and *Eucalyptus*.

### 4.3 Methanotrophic activity and abundance

Oxidizing $CH_4$ from the atmosphere is an important forest ecosystem function (Tate, 2015). It is well known that soil $CH_4$ uptake is mediated by a specific group of soil bacteria (i.e., methane-oxidizing bacteria (methanotrophs)), that can only use $CH_4$ as their sole source of C and energy, with only a few exceptions (Hanson and Hanson, 1996). Soil methanotrophs can be divided into two groups (low- and. high-affinity methanotroph), based on differences in their phylogenetic affiliation, C

assimilation pathway and phospholipid fatty acid composition (Hanson and Hanson, 1996). In forest soils, methanotrophs are known to be dominated by high-affinity methanotrophs such as *Methylosinus* spp. (Kolb, 2009). In our study, the higher *Methylosinus* abundances under slash pine and *Eucalyptus* may be responsible for the higher potential methanotrophic activities (Fig. 6). This is most likely to represent the naturally occurring process of microbial $CH_4$ oxidation under field conditions because high-throughput sequencing of the total microbial communities is unlikely to favor the detection of a





certain functional group. The high frequency of *Methylosinus* thus suggests these microorganisms have been well adapted to $CH_4$-depleted environments and played an important role in atmospheric $CH_4$ oxidation. However, the as-yet-cultivated USCα and USCγ genera are also considered to be an important high-affinity methanotrophs in forest soils (Levine et al., 2011). These microorganisms can be detected only through their *pmoA* genes, which encode for an active subunit of the key

enzyme of particulate $CH_4$ monooxygenase. Comparative analysis of both 16S rRNA and *pmoA* genes would provide useful insights into the adaptation of microbially-mediated $CH_4$ oxidation to long-term plantation regimes in forested soils.

## 5 Conclusions

Through comparing the community structure and diversity of soil bacteria and eukaryotes in soils under different tree species using high-throughput sequencing, we found that slash pine and *Eucalyptus* formed a cluster of soil bacterial and eukaryotic communities that were clearly different from the clusters of hoop pine and kauri pine. Soil pH and nutrient quality indicators such as C:N and EOC:EON ratios were key factors determining the patterns of soil bacterial and eukaryotic communities among the different tree species treatments. Slash pine and *Eucalyptus* had lower soil bacterial and eukaryotic OTU numbers

and diversity indices than hoop pine and kauri pine. We introduced a key factor limitation hypothesis to explain this phenomenon and this hypothesis can give a reasonable explanation for the lower diversity index under slash pine and *Eucalyptus*. Slash pine and *Eucalyptus* had a higher soil $CH_4$ oxidation capacity than the other tree species treatments. Overall, our results suggested that long-term plantations of different tree species can significantly alter soil microbial community structure via changes in soil pH and nutrient quality, thus resulting in differences in soil microbial diversity and

ecosystem functioning.

**Acknowledgements.**We thank Prof. Gary Bacon, Dr. Haibo Dong, Dr. Zhongming Lan for their assistance in soil sampling. This study was supported by East China Normal University (No. 13903-120215-10041), the National Natural Science Foundation of China (No. 31600406) and the Griffith University Research Fellowship.

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



**Table 1.** Soil physiochemical properties in 78-year-old forest plantations with different tree species.

| Properties | Slash pine | Hoop pine | Kauri pine | *Eucalyptus* |
|---|---|---|---|---|
| **Moisture (%)** | 4.26±0.22b | 3.11±0.53b | 3.09±0.67b | 7.69±1.66a |
| **pH** | 4.58±0.03b | 5.64±0.22a | 6.01±0.23a | 4.49±0.04b |
| **Total C (g kg$^{-1}$)** | 13.81±0.81b | 10.13±1.52b | 8.92±1.57b | 26.11±4.37a |
| **Total N (g kg$^{-1}$)** | 0.44±0.03b | 0.43±0.05b | 0.42±0.09b | 0.87±0.14a |
| **C:N** | 31.8±0.7a | 23.1±1.3b | 21.2±0.7b | 29.8±0.6a |
| **EOC (mg kg$^{-1}$)** | 340±41b | 341±31b | 360±30b | 625±77a |
| **EON (mg kg$^{-1}$)** | 14.7±2.9b | 18.4±1.9ab | 23.1±1.5a | 22.4±1.8a |
| **EOC:EON** | 24.17±1.81a | 18.79±1.48b | 15.66±0.87b | 27.64±1.64a |
| **$^{15}$N natural abundance** | 2.86±0.23a | 0.75±0.39b | 0.78±0.28b | 1.51±0.12b |

25

C, carbon; N, nitrogen; EOC, extractable organic C; EON, extractable organic N

Different letters in the same row indicate significant differences at *P*<0.05 among tree species.

30



**Table 2.** Selected statistics of the Miseq sequencing output from 78-year-old forest plantations with different tree species.

| Treatment | Slash pine | Hoop pine | Kauri pine | *Eucalyptus* |
|---|---|---|---|---|
| **Total numbers of prokaryotes sequences** | 79665±22306b | 86640±4379b | 83370±3207b | 116776±1812a |
| **Bacteria (%) of total sequences** | 99.73±0.08a | 99.55±0.04b | 99.49±0.09b | 99.75±0.03a |
| **Archaea(%) of total sequences** | 0b | 0b | 0b | 0.02±0.01a |
| **Unclassified (%) of bacterial sequences** | 0.27±0.04b | 0.45±0.04a | 0.51±0.09a | 0.23±0.02b |
| **Total numbers of Eukaryotes** | 57768±10653 | 65719±5317 | 58195±6964 | 64436±5930 |

Different letters indicate significant differences at $P<0.05$ among the treatments.


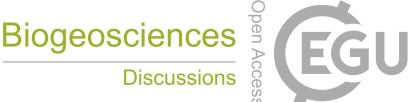

**Table 3.** Estimation of operational taxonomical units (OTUs), Shannon diversity index (H), evenness and chaos scores of soil bacterial and eukaryotes in 78-year-old forest plantations with different tree species.

| | Slash pine | Hoop pine | Kauri pine | *Eucalyptus* |
|---|---|---|---|---|
| **Bacteria** | | | | |
| **OTUs** | 1400±146b | 2486±175a | 2751±151a | 1538±37b |
| **H** | 6.89±0.12b | 8.33±0.38a | 8.92±0.23a | 6.87±0.11b |
| **Evenness** | 0.9747±0.0042b | 0.9883±0.0035a | 0.9937±0.0011a | 0.9768±0.0022b |
| **Chaos** | 1934±140b | 2968±172a | 3211±132a | 2084±79b |
| | | | | |
| **Eukaryotes** | | | | |
| **OTUs** | 528±24c | 792±53ab | 891±46a | 768±8.7b |
| **H** | 4.61±0.19b | 4.84±0.35b | 5.82±0.35a | 5.17±0.21ab |
| **Evenness** | 0.9027±0.0131ab | 0.8838±0.0232b | 0.9392±0.0143a | 0.9089±0.0221ab |
| **Chaos** | 705±29b | 1001±72a | 1069±55a | 932±15ab |

25

Different letters within each row indicate significant differences at *P*<0.05.

30



**Table 4**. Correlation among key physiochemical factors and operational taxonomical units (OTUs) numbers, Shannon diversity index (H) and evenness of soil bacteria and eukaryotes in 78-year-old forest plantations with different tree species.

| r | Bacterial OTUs | Bacterial H | Bacterial evenness | Eukaryotic OTUs | Eukaryotic H | Eukaryotic evenness |
|---|---|---|---|---|---|---|
| **C:N** | -0.951[**] | -0.948[**] | -0.856[**] | -0.755[**] | -0.571[**] | -0.287 |
| **EOC:EON** | -0.831[**] | -0.869[**] | -0.742[**] | -0.49 | -0.446 | -0.261 |
| **pH** | 0.949[**] | 0.962[**] | 0.852[**] | 0.709[**] | 0.619[**] | 0.394 |

[**]indicates significance at $P<0.01$.

C, carbon; EOC, extractable organic carbon; EON, extractable organic nitrogen; N, nitrogen.

25



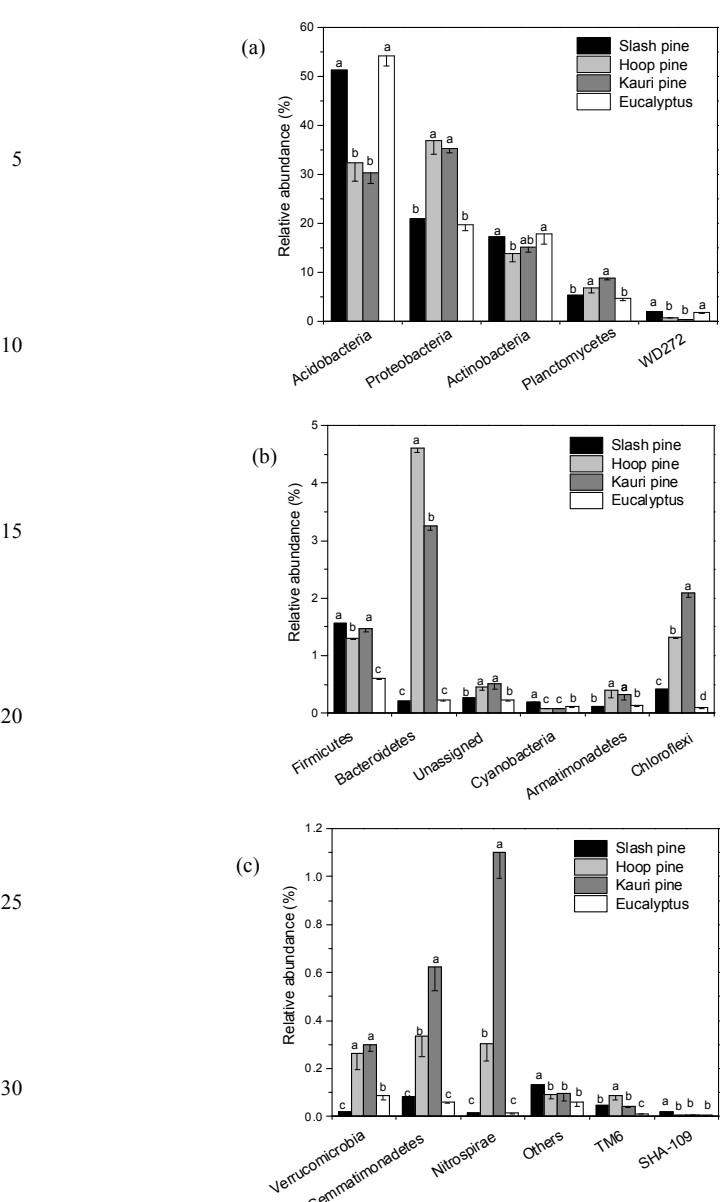

**Figure 1.** Most abundant bacterial phyla in soils under different tree species.





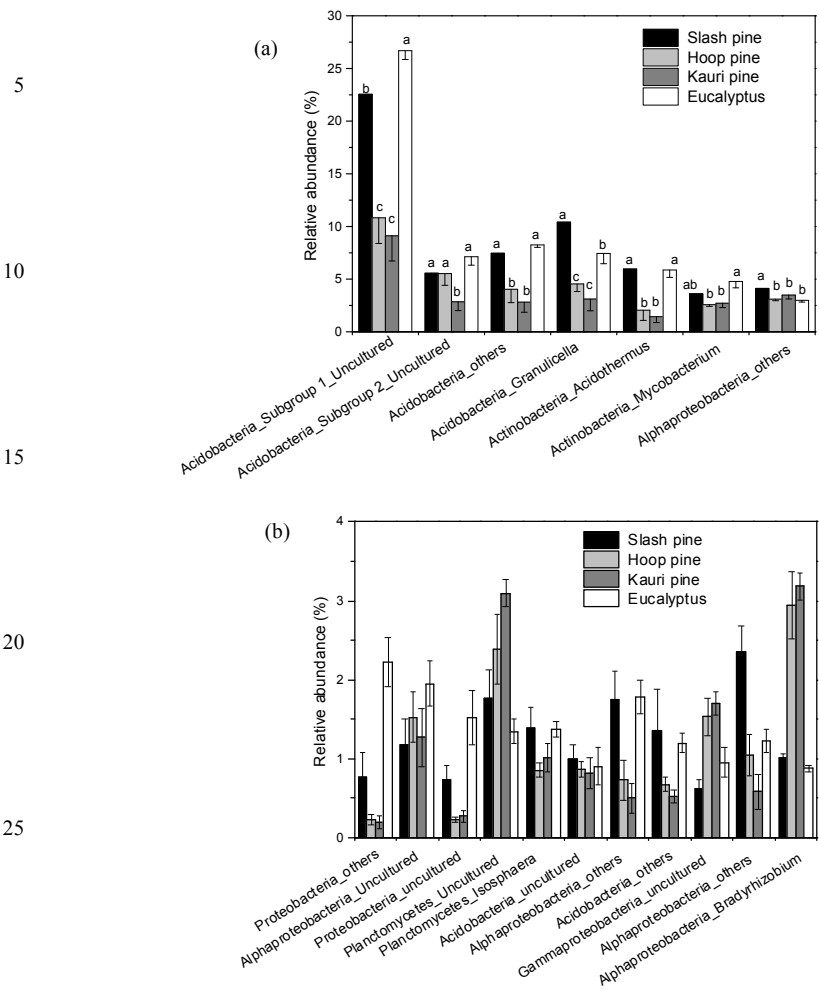

**Figure 2.** Most abundant bacterial genera in soils under different tree species.





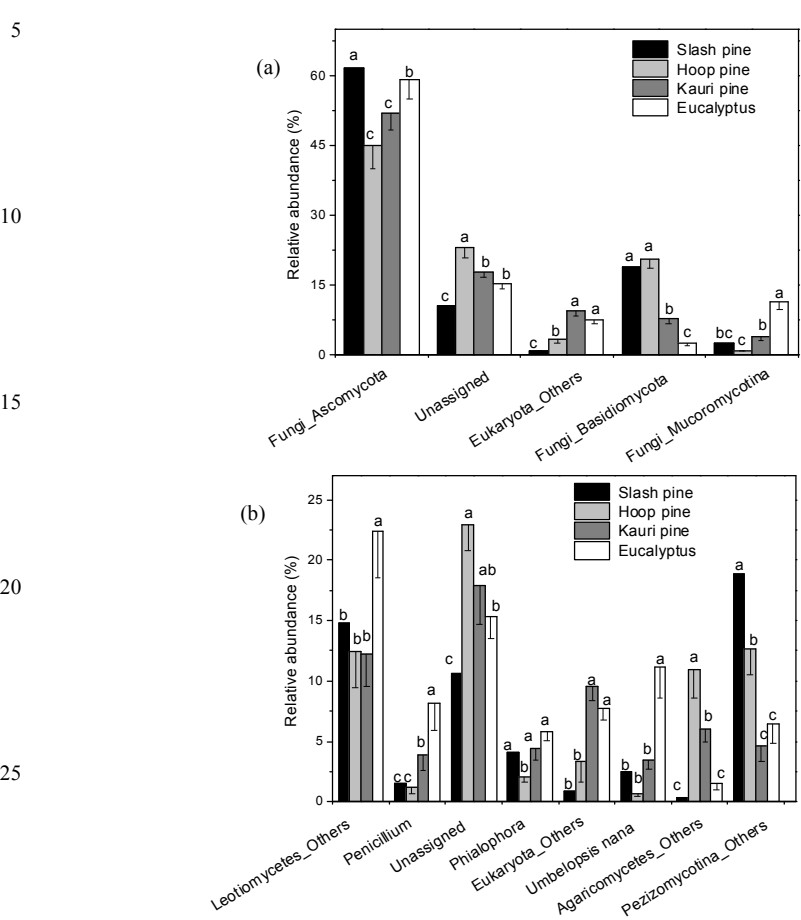

**Figure 3.** Most abundant eukaryotic phyla (a) and genera (b) in soils under different tree species.





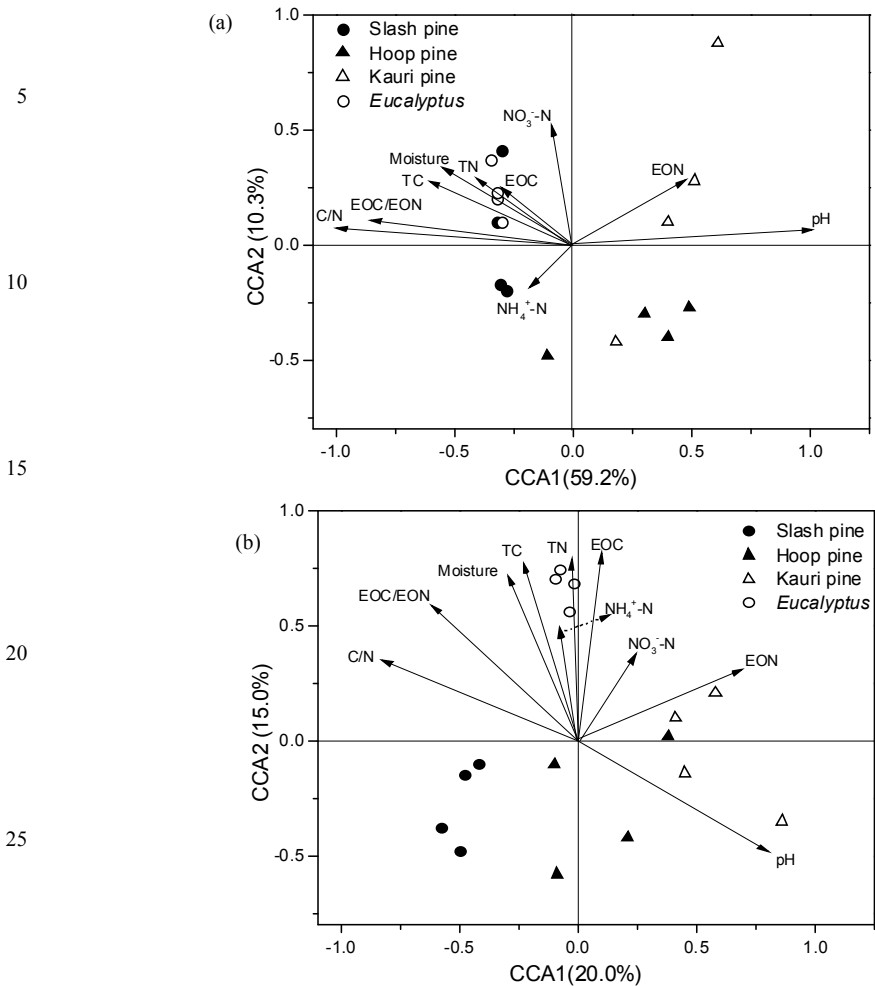

**Figure 4.** Canonical correlation analysis (CCA) of the relationships between soil physiochemical properties and bacterial (a) and eukaryotic (b) communities under different tree species.





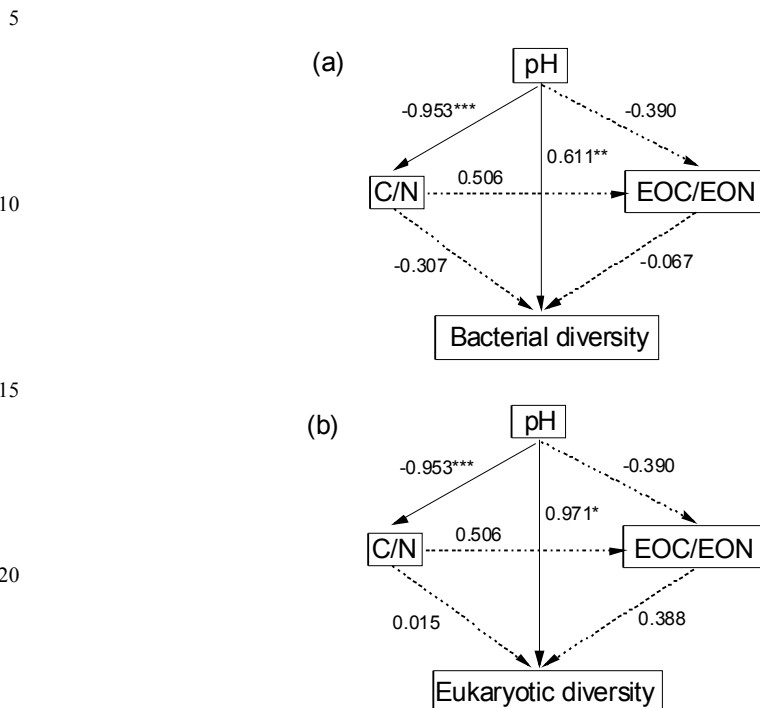

25  **Figure 5.** Path diagrams representing the final model showing the contributions of soil pH and nutrient quality to soil bacterial diversity (a) and eukaryotic diversity (b) under different tree species. Numbers associated with single-headed arrows are partial regression coefficients of multiple regressions. C:N, ratio of soil total C to total N contents; EOC:EON, ratio of soil extractable organic C to extractable organic N contents. Solid arrows denote the directions and effects that were significant ($P<0.05$); the numbers on these pathways are the coefficients. Dashed arrows represent the directions and effects that were not significant (P > 0.05). [*], [**] and [***] indicate significant differences at $P<0.05$, $P<0.01$ and $P<0.001$, respectively.



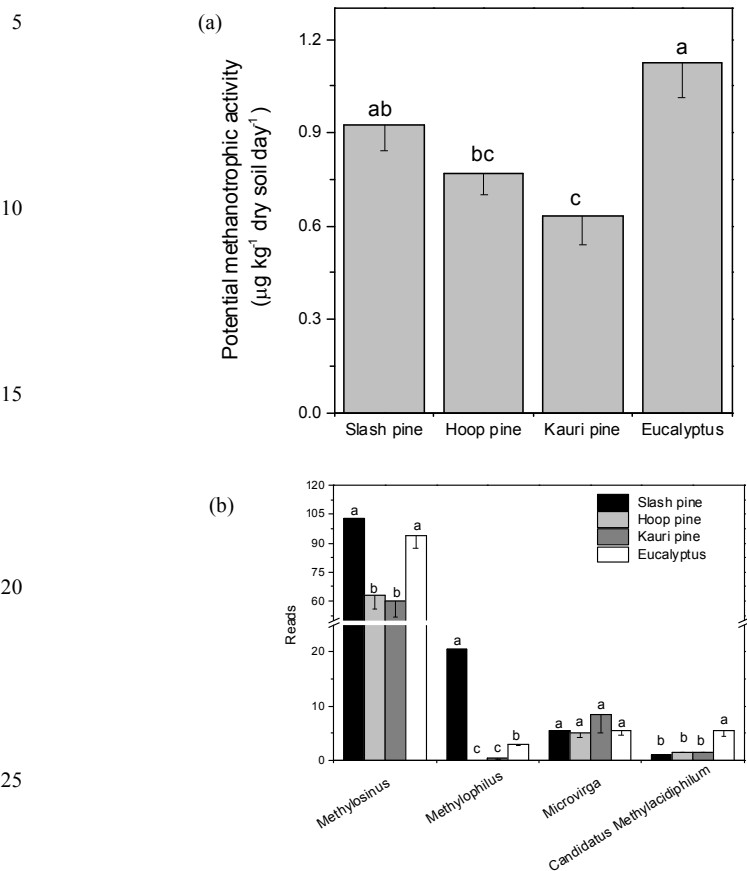

**Figure 6.** Soil potential methanotrophic activity (a) and methylotrophic genus reads detected by high-throughput sequencing (b) under different tree species.