# Peer review of "Soil microbial community structure and diversity are largely influenced by soil pH and nutrient quality in 78-year-old tree plantations"

_Biogeosciences, 2016_

## Referee Comment (RC1) · Anonymous Referee #1 · 19 Feb 2017

This study investigates long-term ecological consequences of forest plantations on the community structure and diversity of soil microorganisms. After initial improvements, the manuscript is well written in a clear way. I suggest that it can be accepted to publish in Biogeosciences after minor revision. Please see my comments as follows:

1. Although the authors have mentioned the role of microorganisms in plantations, the significance for elucidating how forest plantations affect microbial community structure and diversity. Plantations are mainly used to supply with timber. Its sustainability will be a target. It will be good if producing more timber and increasing soil organic matter at the same time. To achieve this, a prerequisite is to know how forest plantations affect microbial community structure and diversity. The reason is that microorganisms drive biogeochemical cycles. Here suggest that the authors develop a frame for microbial function to connect with the aim of this study and clarify its importance. 2. Soil CH4-oxidizing bacteria are one of the important microbial groups, but there are also many others. Why oil CH4 oxidation processes and CH4-oxidizing bacterial communities were identified in this study? The authors should clarify this. 3. In this study, soil $\delta$15N was measured but it seemed that the authors did not mention why measure it. Please clarify what information it can represent. 4. This study focused on soil microbial community structure and diversity. In the text, the authors used soil bacteria and eukaryotes. It is better to briefly mention the main structure and their main functions in somewhere so that the readers are easy to follow. 5. In conclusion, the authors draw a conclusion that soil pH and nutrient quality indicators such as C:N and EOC:EON ratios were key factors determining the patterns of soil bacterial and eukaryotic communities. If it is possible, it will be better to make this change connected with biogeochemical cycles and thus the consequences.
* * *

---

## Referee Comment (RC2) · Anonymous Referee #2 · 28 Feb 2017

Dear authors and editors,

First of all please receive all my apologies for being late to deliver my comments on the new version of the manuscript. Overall, I think this manuscript is of very good quality: it is well written and it addresses good scientific questions. Therefore I recommend to the editors to accept this manuscript after some minor revisions. You will find below my main comments. Sincerely yours,

The authors propose to study the long-term selection of soil microbial communities in different tree plantations developed on the same soil parent material.

1. In the introduction the general context is well explained. The authors proposed

to go further by investigating the soil microbial communities but also their associated crucial function in the context of climate change mitigation (i.e. their CH4-oxidation activity). However, it could be interesting in this perspective to look also to the potential denitrification activity of these communities. Indeed, it is now admitted that NOx can be powerful greenhouse gas specifically in arable lands and where we can observe large NOx emissions. The dataset shows for example that Kauri Pine plots have both the highest EON and the highest relative abundance of the Nitrospirae phylum (known to be implicated in the nitrogen cycle and into the denitrification process). I do not recommend that the authors delay the publication of the manuscript to investigate this question my purpose is to suggest to go further in linking microbial community structure and their functions in future studies. By the way, as I said in the first evaluation of this manuscript, I would recommend to the authors to include in their future studies a plot maintained as arable land. This improvement would allow the authors to distinguish the "afforestation effect" and the "tree species effect" on microbial communities.

2. Mat&Met: I would ask to the authors why did they include 15N data? They do not really use it in the manuscript (by the way, I did not see the 15N arrows in Fig.4). Moreover, the highest values (indicting a processed/old soil organic matter) are found in Slash Pine Pine and Eucalyptus plots where C/N ratios are also the highest (indicating a fresh status of organic matter but it is maybe a tree species effect). Briefly, it is more confusing than informative. Therefore I would suggest to explain in more details the 15N pattern observed or not to mention it at all.

3. Results. In Table 1 I would recommend to the authors to precise the units for soil moisture: what is represented with this % ? the relative volume of water-filled pore space? the relative volume compared to water holding capacity? I would suggest to give more informations on it or to express the data as grams of water per grams of dry soil.

---

## Author Comment (AC1) · 13 Mar 2017

Dear Editor Prof. Yakov Kuzyakov,

We have carefully revised the manuscript according to reviewers' suggestions. The changes we made have been highlighted in yellow throughout the manuscript as shown at the attached file. We hope that it can meet the standard required by your journal Biogeosciences.

Best regards,

Xiaoqi Zhou on behalf of all authors

[Figure]

East China Normal University, Shanghai 200241, China

Reviewer #1 This study investigates long-ter m ecological consequences of forest plantations on the community structure and diversity of soil microorganisms. After initial improvements, the manuscript is well written in a clear way. I suggest that it can be accepted to publish in Biogeosciences after minor revision. R: Thanks.

Please see my comments as follows: 1. Although the authors have mentioned the role of microorganisms in plantations, the significance for elucidating how forest plantations affect microbial community structure and diversity. Plantations are mainly used to supply with timber. Its sustainability will be a target. It will be good if producing more timber and increasing soil organic matC1ter at the same time. To achieve this, a prerequisite is to know how forest plantations affect microbial community structure and diversity. The reason is that microorganisms drive biogeochemical cycles. Here suggest that the authors develop a frame for microbial function to connect with the aim of this study and clarify its importance. R: This has been revised. Lines 13-16 Page 2.

2. Soil $CH_4$-oxidizing bacteria are one of the important microbial groups, but there are also many others. Why oil CH4 oxidation processes and CH4-oxidizing bacterial communities were identified in this study? The authors should clarify this. R: This has been added. Lines 1-4 Page 3.

3. In this study, soil $\delta$15N was measured but it seemed that the authors did not mention why measure it. Please clarify what information it can represent. R: Thanks for your kind comments. This has been added. Lines 1-3 Page 9.

4. This study focused on soil microbial community structure and diversity. In the text, the authors used soil bacteria and eukaryotes. It is better to briefly mention the main structure and their main functions in somewhere so that the readers are easy to follow. R: This has been added. Lines 17-19 Page 3.

5. In conclusion, the authors draw a conclusion that soil pH and nutrient quality

indicators such as C:N and EOC:EON ratios were key factors determining the patterns of soil bacterial and eukaryotic communities. If it is possible, it will be better to make this change connected with biogeochemical cycles and thus the consequences. R: This has been added. Lines 29-31 Page 10.

Please also note the supplement to this comment:
http://www.biogeosciences-discuss.net/bg-2016-552/bg-2016-552-AC1-supplement.pdf

—————————————————

---

## Author Comment (AC2) · 13 Mar 2017

Dear Editor Prof. Yakov Kuzyakov,

We have carefully revised the manuscript according to reviewers' suggestions. The changes we made have been highlighted in yellow throughout the manuscript as shown at the attached file. We hope that it can meet the standard required by your journal Biogeosciences.

Best regards,

Xiaoqi Zhou on behalf of all authors

[Figure]

East China Normal University, Shanghai 200241, China

Reviewer #2 First of all please receive all my apologies for being late to deliver my comments on the new version of the manuscript. Overall, I think this manuscript is of very good quality: it is well written and it addresses good scientific questions. Therefore I recommend to the editors to accept this manuscript after some minor revisions. You will find below my main comments. R: Thanks.

The authors propose to study the long-term selection of soil microbial communities in different tree plantations developed on the same soil parent material. 1. In the introduction the general context is well explained. The authors proposed to go further by investigating the soil microbial communities but also their associated crucial function in the context of climate change mitigation (i.e. their CH4-oxidation activity). However, it could be interesting in this perspective to look also to the potential denitrification activity of these communities. Indeed, it is now admitted that NOx can be powerful greenhouse gas specifically in arable lands and where we can observe large NOx emissions. The dataset shows for example that Kauri Pine plots have both the highest EON and the highest relative abundance of the Nitrospirae phylum (known to be implicated in the nitrogen cycle and into the denitrification process). I do not recommend that the authors delay the publication of the manuscript to investigate this question my purpose is to suggest to go further in linking microbial community structure and their functions in future studies. By the way, as I said in the first evaluation of this manuscript, I would recommend to the authors to include in their future studies a plot maintained as arable land. This improvement would allow the authors to distinguish the "afforestation effect" and the "tree species effect" on microbial communities. R: Thanks for your kind comments.

2. Mat&Met: I would ask to the authors why did they include 15N data? They do not really use it in the manuscript (by the way, I did not see the 15N arrows in Fig.4). Moreover, the highest values (indicting a processed/old soil organic matter) are found in Slash Pine and Eucalyptus plots where C/N ratios are also the highest (indicating a

fresh status of organic matter but it is maybe a tree species effect). Briefly, it is more confusing than informative. Therefore I would suggest to explain in more details the 15N patter n observed or not to mention it at all. R: Thanks. Soil $\delta$15N has been added in Fig. 4. Lines 31-32 Page 21.

3. Results. In Table 1 I would recommend to the authors to precise the units for soil moisture: what is represented with this % ? the relative volume of water-filled pore space? the relative volume compared to water holding capacity? I would suggest to give more information on it or to express the data as grams of water per grams of dry soil. R: This has been revised. Line 11 Page 14.

Please also note the supplement to this comment:
http://www.biogeosciences-discuss.net/bg-2016-552/bg-2016-552-AC2-supplement.pdf
* * *
[Figure]

**Supplement:**

[revised manuscript text omitted]

---

## Author Response (AR1)

Dear Editor Prof. Yakov Kuzyakov,

Thank you very much for your positive feedbacks and for giving us an opportunity to resubmit the revised manuscript. Based on the constructive comments from two reviewers, we strongly improved this manuscript. We hope the revised manuscript will meet the standards of your journal Biogeosciences. Please see our detailed point-by-point responses below.

Best regards,

Xiaoqi Zhou on behalf of all authors

East China Normal University, Shanghai 200241, China

**Reviewer #1**

This study investigates long-term ecological consequences of forest plantations on the community structure and diversity of soil microorganisms. After initial improvements, the manuscript is well written in a clear way. I suggest that it can be accepted to publish in Biogeosciences after minor revision.

R: Thanks for your positive comments.

Please see my comments as follows:

1. Although the authors have mentioned the role of microorganisms in plantations, the significance for elucidating how forest plantations affect microbial community structure and diversity. Plantations are mainly used to supply with timber. Its sustainability will be a target. It will be good if producing more timber and increasing soil organic matter at the same time. To achieve this, a prerequisite is to know how

forest plantations affect microbial community structure and diversity. The reason is that microorganisms drive biogeochemical cycles. Here suggest that the authors develop a frame for microbial function to connect with the aim of this study and clarify its importance.

R: Thanks for your kind suggestions.

Afforestation with different tree species has been accepted as an effective measure for increasing soil C stocks. However, considering the demand for timber of certain tree species and to optimize the productivity of stemwood, forest farmers usually select certain tree species such as coniferous tree species (Lu et al., 2012). The effects of afforestation with different tree species on soil C stocks have been widely studied (Nave et al., 2013; Vesterdal et al., 2013), showing that different tree species can greatly affect soil C sequestration (Vesterdal et al., 2013). On the other hand, the sustainability of afforestation has received wide attention. It is necessary to investigate the effects of afforestation with different tree species on soil microbial diversity and ecosystem functions such as methane (CH4) oxidation capacity, as these functions drive soil biogeochemical cycling. See lines 9-16 page 2.

2. Soil CH4-oxidizing bacteria are one of the important microbial groups, but there are also many others. Why soil CH4 oxidation processes and CH4-oxidizing bacterial communities were identified in this study? The authors should clarify this.

R: Thanks for kind suggestions.

We first describe 'It is necessary to investigate the effects of afforestation with different tree species on soil microbial diversity and ecosystem functions such as methane (CH4) oxidation capacity, as these functions drive soil biogeochemical cycling. 'See lines 14-16 page 2.

Secondly, atmospheric CH4 is the second most important greenhouse gas after CO2, contributing about 25% to global warming, and CH4 has about 25–30 times warming potential of CO2 on a molecular basis. Therefore, CH4 productions and oxidation in soils have received wide attention (Stocker et al., 2013). Upland forest soils mostly function as significant CH4 sinks (Kolb, 2009), but CH4 oxidation capacity in these

soils is sensitive to land uses such as afforestation with different tree species (Tate, 2015). Soil  $CH_4$ -oxidizing bacteria are considered to be the sole microorganisms responsible for consumption of  $CH_4$  in the atmosphere by forest soils (Knief et al., 2005). However, until now little is known about the long-term effects of different tree species on soil potential methane uptake and associated methane-oxidizing bacterial communities. See lines 1-11 page 3.

3. In this study, soil  $\delta^{15}$ N was measured but it seemed that the authors did not mention why measure it. Please clarify what information it can represent.

R:Thanks for your kind suggestions.

We acknowledge that we did not mention soil  $\delta^{15}N$  in the text. In fact, soil  $\delta^{15}N$  reflects net N cycling processes as influenced by tree species, which is affected by atmospheric N inputs and soil N cycling such as nitrification/denitrification rates. However, we did not measure soil nitrification/denitrification rates, so after many considerations and according to the suggestions of reviewer 2, we finally decide to delete soil  $\delta^{15}N$  in the manuscript.

4. This study focused on soil microbial community structure and diversity. In the text, the authors used soil bacteria and eukaryotes. It is better to briefly mention the main structure and their main functions in somewhere so that the readers are easy to follow.

**R: Thanks for your kind suggestions.**

Previous studies have shown that the composition of fungal and bacterial communities in forest soils is largely determined by the tree plantation (Urbanová et al., 2015). On the other hand, microbial communities are responsible for organic matter mineralization and providing nutrients for tree growth, and are thus an integral component of global C and N cycles. Generally, soil with higher C:N ratios under different tree species has more recalcitrant biopolymers, which are represented by the polysaccharides and cellulose that are used by decomposer microorganisms. Because of their filamentous form, fungi tend to be more involved in the decomposition of polymeric compounds (de Boer et al., 2005). Bacteria that preferentially use organic

compounds with a low molecular mass may rely on the products of fungal-biopolymer decomposition for nutrition (Tedersoo et al., 2008). To investigate the effects of long-term tree species plantations on soil microbial communities, we used bacterial-specific primers and eukaryotic-specific primers to investigate changes in the community structure and diversity of soil bacteria and fungi, respectively, in response to long-term afforestation with different tree species. See lines 21-29 page 2 and lines 18-21 page 3.

5. In conclusion, the authors draw a conclusion that soil pH and nutrient quality indicators such as C:N and EOC:EON ratios were key factors determining the patterns of soil bacterial and eukaryotic communities. If it is possible, it will be better to make this change connected with biogeochemical cycles and thus the consequences.

R: Apart from the conclusion that soil pH and nutrient quality indicators were key factors determining the patterns of soil bacterial and eukaryotic communities, we also found that sash pine and Eucalyptus had a higher soil  $CH_4$  oxidation capacity than the other tree species treatments. Overall, our results suggested that long-term plantations of different tree species can significantly alter soil microbial community structure via changes in soil pH and nutrient quality, thus resulting in differences in soil microbial diversity and ecosystem functioning. On the other hand, our results highlight the importance of soil acidification, which outweighed nutrient quality and played a predominant role in soil microbial community patterns among the tree species, leading to lower microbial diversity under slash pine and Eucalyptus. See lines 24-29 page 10.

**Reviewer #2**

First of all please receive all my apologies for being late to deliver my comments on the new version of the manuscript. Overall, I think this manuscript is of very good quality: it is well written and it addresses good scientific questions. Therefore I recommend to the editors to accept this manuscript after some minor revisions. You will find below my main comments.

R: Thanks for your positive comments.

The authors propose to study the long-term selection of soil microbial communities in different tree plantations developed on the same soil parent material.

1. In the introduction the general context is well explained. The authors proposed to go further by investigating the soil microbial communities but also their associated crucial function in the context of climate change mitigation (i.e. their CH4-oxidation activity). However, it could be interesting in this perspective to look also to the potential denitrification activity of these communities. Indeed, it is now admitted that NOx can be powerful greenhouse gas specifically in arable lands and where we can observe large NOx emissions. The dataset shows for example that Kauri Pine plots have both the highest EON and the highest relative abundance of the Nitrospirae phylum (known to be implicated in the nitrogen cycle and into the denitrification process). I do not recommend that the authors delay the publication of the manuscript to investigate this question my purpose is to suggest to go further in linking microbial community structure and their functions in future studies. By the way, as I said in the first evaluation of this manuscript, I would recommend to the authors to include in their future studies a plot maintained as arable land. This improvement would allow the authors to distinguish the "afforestation effect" and the "tree species effect" on microbial communities.

R: Thanks for your kind comments. You are right. We are very regretful that we cannot find a continuous cultivation site as a control site. As it is, we cannot distinguish the "afforestation effect" and the "tree species effect" on microbial communities as mentioned by reviewer. In this study, we mainly focus on the effects of long-term tree species plantations on soil microbial community structure and diversity. We acknowledge that we did not measure soil nitrification/denitrification rates, so we did not mention too much about N cycling and underlying microbial processes. In the future, we will plan to investigate the effects of long-term extreme drought events on soil N mineralization rates, nitrification/denitrification rates and

underlying microbial mechanisms in a subtropical evergreen forest. We hope that we can get a good result.

2. Mat&Met: I would ask to the authors why did they include 15N data? They do not really use it in the manuscript (by the way, I did not see the 15N arrows in Fig.4). Moreover, the highest values (indicting a processed/old soil organic matter) are found in Slash Pine and Eucalyptus plots where C/N ratios are also the highest (indicating a fresh status of organic matter but it is maybe a tree species effect). Briefly, it is more confusing than informative. Therefore I would suggest to explain in more details the 15N patter n observed or not to mention it at all.

R: Thanks for your kind suggestions.

We acknowledge that we did not mention soil  $\delta^{15}N$  in the text. In fact, soil  $\delta^{15}N$  reflects net N cycling processes as influenced by tree species, which is affected by atmospheric N inputs and soil N cycling such as nitrification/denitrification rates. However, we did not measure soil nitrification/denitrification rates, so after many considerations and according to the suggestions of reviewer 2, we finally decide to delete soil  $\delta^{15}N$  in the manuscript.

3. Results. In Table 1 I would recommend to the authors to precise the units for soil moisture: what is represented with this % ? the relative volume of water-filled pore space? the relative volume compared to water holding capacity? I would suggest to give more information on it or to express the data as grams of water per grams of dry soil.

R: Done. The unit of soil moisture has been revised into  $g kg^{-1} dry$  soil. The revised Table 1 has been shown below. See line 11 page 14.

| Properties                              | Slash pine  | Hoop pine   | Kauri pine  | Eucalyptus  |
|-----------------------------------------|-------------|-------------|-------------|-------------|
| <mark>Moisture (g kg⁻¹ dry soil)</mark> | 42.6±2.2b   | 31.1±5.3b   | 30.9±6.7b   | 76.9±16.6a  |
| рН                                      | 4.58±0.03b  | 5.64±0.22a  | 6.01±0.23a  | 4.49±0.04b  |
| Total C (g kg -1 )           | 13.81±0.81b | 10.13±1.52b | 8.92±1.57b  | 26.11±4.37a |
| Total N (g kg -1 )           | 0.44±0.03b  | 0.43±0.05b  | 0.42±0.09b  | 0.87±0.14a  |
| C:N                                     | 31.8±0.7a   | 23.1±1.3b   | 21.2±0.7b   | 29.8±0.6a   |
| EOC (mg kg -1 )              | 340±41b     | 341±31b     | 360±30b     | 625±77a     |
| EON (mg kg -1 )              | 14.7±2.9b   | 18.4±1.9ab  | 23.1±1.5a   | 22.4±1.8a   |
| EOC:EON                                 | 24.17±1.81a | 18.79±1.48b | 15.66±0.87b | 27.64±1.64a |

Table 1. Soil physiochemical properties in 78-year-old forest plantations with different tree species.

C, carbon; N, nitrogen; EOC, extractable organic C; EON, extractable organic N Different letters in the same row indicate significant differences at *P*<0.05 among tree species.